# Word Predictability is Based on Context – and/or Frequency

## Abstract

In this paper we present an experiment carried out with BERT on a small number of Italian sentences taken from two domains: newspapers and poetry domain. They represent two levels of increasing difficulty in the possibility to predict the masked word that we intended to test. The experiment is organized on the hypothesis of increasing difficulty in predictability at the three levels of linguistic complexity that we intend to monitor: lexical, syntactic and semantic level. Whereas lexical predictability may be based on word frequency and not just context, syntax and semantics strictly constrain meaning understanding. To test this hypothesis we alternate canonical and non-canonical version of the same sentence before processing them with the same DL model. In particular, we expect the poetry domain to introduce additional restrictions on the local word context due to the need to create metaphors which require non-literal meaning compositional processes. The result shows that DL models are highly sensitive to presence of non-canonical structures and to local non-literal meaning compositional effect. However, DL are also very sensitive to word frequency by predicting preferentially function vs content words, collocates vs infrequent word phrases. To measure differences in performance we created a linguisticalluy based "predictability parameter" which is highly correlated with a cosine based classification but produces better distinctions between classes.

## 1 Introduction

In this paper we will discuss in detail the set up and results of an experiment carried out with a small dataset of Italian sentences, using the output of the first projection layer of a Deep Learning model, the raw word embeddings. We decided to work on Italian to highlight its difference from English in an extended number of relevant linguistic properties. The underlying hypothesis aims at proving the ability of BERT(Ashish Vaswani and Polosukhin, 2017) to predict masked words with increasing complex contexts. To verify this hypothesis we selected sentences that exhibit two important features of Italian texts, non-canonicity and presence of words with very low or rare frequency. To better evaluate the impact of these two factors on word predictability we created a word predictability measure which is based on a combination of scoring functions for context and word frequency of (co-)occurrence. The experiment uses BERT assuming that DNNs can be regarded capable of modeling the behaviour of the human brain in predicting a next word given a sentence and text corpus - but see the following section.

It is usually the case that paradigmatic and syntagmatic properties of words in a sentence are tested separately. In this experiment we decided to test them together by combining non-canonicity and infrequent word choice. Italian sentences are taken from two domains: newspapers and poetry domain. Italian bureaucratic and newspaper language can be easily understood by sufficiently literate people. This is not so for people affected by return illiteracy, which constitute a good majority of middle aged people. The second set of sentences taken from Italian poetry of last century is fairly hard to understand. This is due both to choice of infrequent words and uncommon structures. The hypothesis is that predictable masked words would be more frequent in the first than in the second set of sentences. In addition we expect the canonical version of the dataset to be more predictable.

We decided to work with a small dataset which is made of 18 sentences with 150 content words which are then duplicated in the canonical structure thus summing up to 36 sentences and 300 words. This has been done in order to be able to comment the import of every single masked word and its role in the overall sentence structure from a linguistic point of view. This has allowed us to come to precise conclusions on the type of errors

the encoding phase systematically makes. In particular, the experiment has allowed us to evaluate the bias of the model towards one of the domains, the newswire one, where the best results have been obtained. In the case of the poetry domain, errors where in general mostly due to the absence of word embeddings with an appropriate context for the input word and the consequent inability to predict the masked word.

The most important feature of the experiment is that all sentences are characterized by non-canonical structures. Italian is a language in which non-canonical structures are fairly common due to the weakly configurational nature of the language and to the existence of the pro-drop parameter that allows sentences to freely omit lexically expressed subjects(Delmonte et al., 2007). We then operated on the dataset in two ways: at first we reformulated the text obtained modifying each sentence structure in order to make it canonical. The inclusion of sentences from poetry has been done in order to focus on the effects of context in conjunction with word level frequency effects [1]. The reason for this choice is that poetry is the only domain where rare words are used consistently thus making available a full real context of use for (very) low frequency words. The combined effect of using rare words in a non-canonical syntactic configuration and then restructuring the same sentence with a canonical structure allowed us to make important comparisons. Non-canonical sentences in Italian can be found in great number due to the pro-drop nature of the language which thus resembles Chinese and Japanese (Delmonte, 2009). In addition, Italian is a morphologically rich language thus possessing a very large vocabulary of unique wordforms which, if compared to the total number of wordforms obtainable from the WordNet list of citation forms for English is an order of magnitude higher – from 500K to 5 million wordforms in Italian, only considering the corresponding number of grammatical categories (Delmonte, 2014). We already discussed elsewhere (Delmonte, 2021) that languages like Italian, which have a rich morphology, need embeddings with higher dimensions and a vocabulary size more than doubled in order to account for the variety of semantically relevant wordforms.

When referring to context in BERT, the whole preceding sentence portion is included. BERT be-

ing bidirectional the context will apply to both the right and the left previous sequence of tokens. However, when referred to Distributional Semantic Models, the context is usually determined by the number (2 to 5) of co-occurring tokens to be considered when building vectors for word embeddings: if the masked word is the first word in the sentence only the right context will be available and this fact reduces the ability of prediction as shown by our data. The result of our experiment shows that DNNs are very sensitive to context and that frequency of occurrence is less relevant for word predictability. The paper is organized as follows: in the following section, we introduce briefly state of the art on the problem of word predictability as seen from the cognitive point of view; in section three we present the experimental setup and the typology of non-canonical structures contained in our dataset; section 4 presents the experimental results and discuss its import for the predictability parameter, then our conclusion. In the Appendix we reported the translated version of the sentences, while the detailed analysis is contained in the Supplemental Material.

## 2 Word Predictability in Cognitive and Psycholinguistic Research

Word prediction or predictive language processing has been a foundational topic for psycholinguistic research in the last 50 years or so for all that concerns human sentence processing and comprehension. In this paper we intend to exploit the hypothesis presented lately in a number of papers (Goodkind and Bicknell, 2018) (Schrimpf et al., 2021) where human word predictivity is compared and tested by the performance of DNNs in next-word-prediction tasks. In particular, in their conclusion, Schrimpf et al. comment on the results of their findings defining them as an attempt to create a viable hypothesis for modeling predictive language processing in human brain by the use of predictive artificial neural networks, specifying that so-called "transformer" models – BERT - are best-performing models. In another paper (see (Fedorenko et al., 2020)), had already come to the conclusion that it is by the use of working memory as a whole that word predictivity works: i.e. the integration of all levels of language processing, lexico-semantic, syntax and knowledge of the world conspire to make word prediction viable in order to carry out the primary function of human

---

[1] For a thorough syntactic and semantic description of these sentences, (Delmonte, 2018)

language, "the extraction of meaning from spoken, written or signed words and sentences"(see (Schrimpf et al., 2021),p.2).

The question of word frequency and their predictability is dealt with in great detail in a paper by (N. and Levy, 2013). Words which have high predictability scores are also those which are somehow more related to the prior context, and words which are more related to the prior context are also easier to integrate semantically. "... there is no such thing as an unexpected word; there are only words which are more or less expected." (ibid. 309). In this approach, predictability changes from one word to the next due to syntactic and semantic constraints, eventually coming to the conclusion that speakers tend to choose words more likely to occur in a given context.

Estimating the level of difficulty or the "surprisal" – or unpredictability - of a word in a given context is done by the negative log probability measure which counts as 1 words fully predictable and as 0 those unpredictable, where the former ones convey no additional information as opposed to the latter. Thus, in a serial-search model imagining lexical access in a frequency sorted lexicon, the 100th most frequent word would take twice as long to access as the 50th most frequent word. As a consequence, most frequent words are less informative and are easier to pronounce and to understand. However, this may only be regarded as a theoretically viable hypothesis since even when words are infrequent and unknown they may still serve to formulate some meaning related bit of information and help in understanding the content of the utterance. From the results obtained in our experiment based on BERT raw embeddings, both frequency and context conjure to establish word predictability. In some cases it is clearly the low frequency to prevent embeddings to be made available, but in other cases - see the example of the ambiguous word "ora"/now-hour below - even though the word and the local context is fairly typical, the word is not predicted. A partly similar approach has been attempted by Pedinotti et al.(Rambelli et al., 2021), in a paper where they explore the ability of Transformer Models to predict transitive verb complements in typical predicate-argument contexts. Their results show clearly the inability to predict low frequency near synonyms, thus confirming the sensitivity of BERT-like models to frequency values. The experiment also included a

version of the dataset where the surface syntactic structure of the sentences was modified in order to introduce non-canonical structures. In fact this was only limited, though, to two cases: interrogative and cleft-structures. The second structure showed how the model suffered from non-recurrent word order by an important drop in performance (from 70 to 38% accuracy).

Another parameter which has loomed large in the cognitive literature is the relevance of the effort/time required to pronounce/read a word: a short word, both phonetically and as grapheme, is preferred and confirmed in an experiment based on semantic grounds by Mahowald et al. (Mahowald et al., 2012), where pairs of near synonym words inserted in frame sentences and user have consistently chosen the shortest ones as the most predictable. This seems to be confirmed by the well-known fact that the top range of frequency lists of wordforms are occupied by short words thus confirming the inverse correlation existing between word length and frequency. Most frequent words are not only the shortest but the ones with more senses as confirmed in a paper by Piantadosi et al. (Piantadosi et al., 2012), hence the more frequent. To verify this we inspected the top 200 words in the frequency lists of ItWac for Italian and English and counted their number of syllables with the following results: Italian has 75 monosyllabic words and 125 words with more than one syllable; English has 149 monosyllabic words and 51 words with more syllables. The two languages have an opposite distribution as has also been documented in a previous paper (Delmonte, 2014). In addition, English top 200 words contain only 30 content words, while Italian contains 61 content words, ten of which are morphological variants, English has only one morphological variant.

## 3 The Experimental Setup

We assume that word predictability can be characterized by two parameters: word (co-occurrence) frequency/ies and linguistic complexity measured by syntactic/semantic related scoring functions. We evaluate word co-occurrence frequencies by means of embeddings as the cosine value made available by BERT[2] in its first projection layer, using pre-trained models and no fine-tuning. We used BERT – with the Italian model taken from UWAC corpus,

---

[2] presented in the paper by Loreto Parisi et al. (Parisi et al., 2020)

Umberto-commoncrawl - and examined the output of the first or projection layer[3]. In this way we intended to check the predicting ability of BERT on the masked word, by selecting in turn one content word at a time allowing BERT to use the rest of the sentence as a context to make appropriate predictions.[4] To this aim we ran BERT by masking each content word and some function word, one at a time in order to be able to make a detailed error analysis and parameter evaluation.

The text we use in the experiment has been organized in order be able to focus on the context: it is made up of 18 sentences, 11 belonging to the newswire domain and 7 sentences belonging to Italian poetry of last century[5]. In a section below are the description of the non-canonical features of the sentences we used for the experiment. The English translation is available in the Appendix. We signed every sentence with letter A for those belonging to the poetry domain - 7, and letter B for newswire domain - 11. The newswire sentences are taken from the treebank of Italian – VIT, Venice Italian Treebank – available also under UD repositories[6]; the poetry set of sentences is taken from publicly available collections of poets of the first half of the nineteenth century which have already undergone specific analysis in previous work [7].

In order to evaluate frequency values associated to each masked word, we cleaned the frequency list of Italian wordforms compiled on the basis of ItWaC [8], deleting all numbers and websites. Then we created a list of 50000 most frequent word-forms to be used to check what words would be included by a model created on the basis of BERT tokenization module. Wordforms included are up to a frequency value of 1377. The remaining list is cut at frequency value 4, thus leaving out Rare words, made up of Trislegomena, Dislegomena and Hapaxlegomena, which is by far the longest list: it counts 1,642,949 entries. The Upper List – the list that includes the 50000 plus the rest of wordforms down to and inlcuding words with frequency 4, is made up of 513,427 entries.

Then, we divided the 50000 vocabulary into four subparts on a frequency basis: Subpart 1 the highest, Subpart2 high, Subpart 3 middle and Subpart 4 the lowest. However, to make comparisons easier, we recast this subdivision into two halves: first half with "high" frequency words, including three segments - highest, high and middle frequency words down to 10000 -, second half from 10000 to 1377 we call "low" frequency words. We then consider as "very-low" frequency words those included in the so-called Upper List - from 1377 down to 4 occurrences -, and the remaining long tail are classified as "Rare Words". The final classification is then organized into four classes: High, Low, Very Low and Rare. To make frequencies more visible, we mark with one asterisk words belonging to "Low", with two asterisks words belonging to "Very-Low", and three asterisks "Rare" words.

## 3.1 The Dataset and Non-Canonical Structures

As said above, Italian is very rich in number and types of non-canonical structures. This is also due to its being a direct derivation from Latin, a free word-order language (see (Delmonte, 2018)). Our approach has been previously adopted by other researchers but with slightly different aims that we describe in what follows. The first work is by Paccosi et al.(Paccosi et al., 2022) where the authors present a new dataset of Italian based on "marked" sentences, which is then used to verify the performance of a neural parser of Italian (TINT) on the dataset. The result for LAS dependency structures is 77%, 3 points below the best results previously obtained on the UD corpus of Italian, which was 80% accuracy. This result confirm previous

---

[3]We produced the whole experiment leveraging the ability of the Huggingface implementation (Wolf et al., 2019)

[4]Of course, we are aware of the fact that by training a DNN, its error rate is reduced in cycles through back propagation. This involves comparing its predicted function value to the training data. This is done by computing the gradient of a cross entropy loss error function and proceeding by specified increments of the weights to an estimated optimal level, determined by stochastic gradient descent, which in the case of a test set, does not necessarily correspond to what has been learnt.

Words are represented in a DNN by vectors of real numbers. Each element of the vector expresses a distributional feature of the word - in our case by cosine values. These features are the dimensions of the vectors, and they encode their co-occurrence patterns with other words in a training corpus. Word embeddings are generally compressed into low dimensional vectors (200-300 dimensions) that express similarity and proximity relations among the words in the vocabulary of a DNN model.

[5]We comment and analyze in depth all sentences in a paper where parsers of Italian have been used to parse them and have resulted in an accuracy lower than 50%. (see (Delmonte, 2018))

[6]https://universaldependencies.org/

[7]see (Delmonte et al., 2007) (Delmonte, 2009)

[8]The corpus contains approximately 388,000 documents from 1,067 different websites, for a total of about 250M tokens. All documents contained in the PAISA' corpus date back to Sept./Oct. 2010. The itWaC corpus is available at https://wacky.sslmit.unibo.it/ accessed on October, 2021

work documented also in (Delmonte, 2016) with a small dataset containing strongly marked sentences, which have been included in the text used in this paper, where the results were well below 50% accuracy. The authors make a detailed description of the type of marked structures they annotated in their treebank corpus. It is a list of seven structures - cleft, left dislocated, right dislocated, presentative "ci", inverted subject, pseudo-clefts, hanging topic - with a majority of Cleft sentences and Left dislocated sentences. Similar results are obtained by the experiment presented in the paper by Pedinotti et al. (Rambelli et al., 2021) where in Section IV they test the ability of Transformers - they use RoBERTa - on a small dataset with surface syntactic structures different from the recurrent word order. They modify the sentences to produce cleft and interrogative versions of the same sentences. The result for core semantic roles - this is what they are testing - is a dramatic drop of performance from 0.65 of correlation in canonical transitive versions down below 0.35.

When compared to the corpuses above, our dataset is smaller but it contains many more types of marked constructions, which makes it more difficult to come to terms with, and this is due mainly to presence of sentences from the poetry domain. We present now the structures contained in our dataset:
***complete argument inversion*** (the complement is fronted and the subject is in post verbal position) in sentence 7B - with copula deletion, and in sentence 17B with infinitival structure as subject;
***object fronting*** (the object comes before the subject at the beginning of the sentence) in sentence 2A and 5A;
***adjective extraction*** (the adjective is extracted and fronted from the noun phrase) in sentence 13A and 14A;
***PPadjunct preposing from participial clause*** in sentence 1B and 13A;
***lexical verb left extraction*** (the main verb - untensed non-finite - is positioned before the auxiliary/modal) in sentence 3A;
***subject right dislocation*** (the subject is positioned after the complements) in sentence 3A and 6B;
***subject and object fronting*** (the subject comes before the object and both are positioned before the main verb) in sentence 4A and 5A;
***PPspecification extraction from the noun phrase and fronted to the left*** in sentence 5A;
***clitic left dislocation*** in sentence 8B;
***object right dislocation*** (the object is positioned after the indirect object or the adjuncts) in sentence 10B;
***parenthetical insertion*** (a parenthetical is inserted after the subject before the main verb) in sentence 11B and 16B;
***adjective right extraction*** (the adjective is extracted from the noun phrase and positioned after the noun adjuncts) in sentence 11B and 14A;
***PPspecification right stranding*** - the PPof is stranded to the right out of the noun phrase in sentence 14B;
***lexical verb right extraction*** (the main verb - untensed non-finite - is positioned after the complements) in sentence 12A;
***double parenthetical insertions*** (after the subject and after the verb complex and before the complements) in sentence 15B and 16B;
***clitic left dislocation with subject fronted as hanging topic*** in sentence 18B.

## 4 Experimental Results and Discussion

The evaluation has been carried out in three different configurations: on a first configuration, part of the sentences, the last 7 – are withheld with the aim to reduce the overall context at sentence level. This is done both for non-canonical and canonical structures. Then the last 7 sentences are added and the cosine values verified to see if predictions have been modified.

We assume that a better form of evaluation should account for gradable differences between predictions in which the actual word is not found but the ones predicted are very "similar". The word "similar" then will need to be better decomposed into its various linguistic aspects and we have devised a graduality which may be turned into scores according to simple linguistic criteria. Similarity may attain morphological, lexical, grammatical, syntactic, semantic criteria. Thus the more the choices are close to the actual meaning of the expected word, the higher the score will be which we assume will be a real value from 0 to 1. Since the final choice is done on the basis of the theoretical assumptions underlying the Distributional Semantic Model we will call Table 1. accordingly.

We applied the scores reported in the table to the whole set of sentences and computed the results in the two tables below. In Table 2. we evaluate the seven sentences from the poetry domain, and in Table 3. the eleven sentences from the newswire

| Linguistic Category | Feature Type | Score |
|---|---|---|
| Identical | (first position) | 1 |
| Identical | (second position) | 0.99 |
| Identical | (third position) | 0.97 |
| Identical | (fourth position) | 0.95 |
| Same word | different morphology | 0.8 |
| Same word | different grammatical category | 0.7 |
| Hyponym/ Antonym/ Meronym, Synonym | same morphology same grammatical category | 0.6 |
| Hyponym/ Antonym/ Meronym, Synonym | different morphology same grammatical category | 0.5 |
| Hyponym/ Antonym/ Meronym, Synonym | different morphology different grammatical category | 0.4 |
| Different word | same grammatical category same morphology | 0.3 |
| Different word | same grammatical category different morphology | 0.2 |
| Different word | different grammatical category | 0.1 |
| No word | Punctuation - ‹ukn› | 0 |

Table 1: Graded Evaluation Scale for a Linguistically Based Similarity Scoring according to DSM

domain. We computed three main parameters: in column 2, Number of Words masked with respect to total number of tokens; in columns 3 and 4 we list words correctly predicted with the identical corresponding word respectively in the Non Canonical and in the Canonical sentence structure; then in columns 5 and 6 we list the number of words with frequency values respectively Higher and Lower than a given threshold that we established at 10.000 occurrences. We also considered words that don't appear in the 50000 vocabulary and reported them after a slash: we assume their import should be

valued double. Thus for instance, in the Poetry text, we found 5 such words and the total number of Low Frequency Words is increased by 10 points. Finally, in column 7, we reported the result of applying the scoring function described in Table 1.

| Sent. No. | No. Mask. Ws. | Non Can. W.s | Can. Ws. | High Fr. Ws. | Low Fr. Ws. | Ling. Eval. |
|---|---|---|---|---|---|---|
| 2.A | 10/8 | 0 | 3 | 4 | 3/1 | 3.76 |
| 3.A | 14/9 | 3 | 4 | 6 | 3 | 6.04 |
| 4.A | 10/8 | 2 | 2 | 4 | 4 | 3.99 |
| 5.A | 9/6 | 0 | 0 | 4 | 1/2 | 2 |
| 12.A | 11/7 | 1 | 2 | 4 | 1 | 3.49 |
| 13.A | 15/7 | 0 | 0 | 5 | 0/2 | 2.4 |
| 14.A | 14/9 | 1 | 1 | 6 | 3/1 | 3.1 |
| totals | 83/54 | 7 | 12 | 33 | 15/6 =27 | 24.78 |
| ratios | 0.65 | 0.583 | | | 0.818 | 0.4589 |

Table 2: Linguistic Evaluation of Poetry Sentences

| Sent. No. | No. Mask. Ws. | Non Can. W.s | Can. Ws. | High Fr. Ws. | Low Fr. Ws. | Ling. Eval. |
|---|---|---|---|---|---|---|
| 1.B | 14/8 | 3 | 5 | 8 | 0 | 5.97 |
| 6.B | 6/5 | 2 | 3 | 5 | 0 | 3.84 |
| 7.B | 5/4 | 0 | 0 | 3 | 1 | 2.4 |
| 8.B | 10/7 | 1 | 2 | 6 | 1 | 2.37 |
| 9.B | 7/4 | 2 | 3 | 4 | 1 | 2.99 |
| 10.B | 12/9 | 1 | 1 | 7 | 2 | 4.79 |
| 11.B | 15/10 | 2 | 4 | 10 | 0 | 6.17 |
| 15.B | 25/10 | 7 | 7 | 8 | 2 | 8.23 |
| 16.B | 22/10 | 4 | 4 | 8 | 2 | 7.2 |
| 17.B | 15/9 | 6 | 6 | 10 | 0 | 7.1 |
| 18.B | 22/10 | 4 | 4 | 9 | 0/1 | 5.7 |
| totals | 153/86 | 31 | 38 | 78 | 9/1=11 | 56.76 |
| ratios | 0.56 | 0.816 | | | 0.141 | 0.66 |

Table 3: Linguistic Evaluation of Newswire Sentences

As can be easily noticed by comparing all parameters, poetry and news have opposite values. Quantities measured in column 2 show how the ratio of masked words is higher in poetry than in the news domain – 0.65 › 0.56 -, the reason being that poetry text makes use of less grammatical or function words, like articles, clitics, prepositions which are highly predictable but are less informative. The first important parameter is the difference in number of masked words identified in Non-Canonical

vs Canonical Sentences, and here again as can be easily noticed the newswire domain has a much higher score than the poetry domain – 0.816 › 0.583. Then the second relevant parameter derived by the proportion of High Frequency words vs Low Frequency words and computed as a ratio between the sum of the absolute number of words plus a doubling of the number of very low frequency words. Here the scores show the opposite relation, Poetry domain has a much higher number of Low Frequency words than Newswire domain – 0.818 › 0.141. Eventually, the linguistic evaluation of every single masked word on the basis of its cosine measure and the graded scoring scale reported in Table 1. Where we see again a much higher overall score for the Newswire than the Poetry domain – 0.66 › 0.4589. The conclusion we can safely draw from these data is that the News domain has a higher linguistically and frequency-based evaluated prediction score: - because it has a much lower number of Low Frequency words - because it has a higher number of predicted words in Non-canonical structures In other words, predictability is both dependent on word frequency and their structural position. One example is highly representative of the interplay between frequency and context and is the word "Ora", an ambiguous word with two homographs-homophones: one meaning "now", an adverbial contained in sentence n. 9 - the newswire domain; and another meaning "hour", a (temporal) noun, contained in sentence n. 5 - the poetry domain. Only the adverbial is predicted in both structural versions. The noun is contained in a sentence belonging to the poetry domain where the overall context is not supportive for that word predictability.

Below, we show an evaluation of each sentence based on the sum of cosine values as reported by BERT associated to the first candidate and compare it to the one we organized in table 1. We also show an evaluation with mixed data, by selecting cosine values associated to identical word form if predicted, else the first candidate, which we call Mixed. The sentences are listed in descending order by the magnitude of the Linguistic Parameter.

Correlation evaluation between our Linguistic Parameter and Cosine values is estimated at 0.8705 as can be gathered also visually from Figure 1. below. Values for B texts are overall higher in both evaluations: the descending trend is however more linear for the linguistic parameters than for the co-

| Sent. No. | Ling. Evaluat. | Mixed Evaluat. | Cosine Evaluat. | No. Mask. Ws. |
|---|---|---|---|---|
| 15.B | 8.23 | 3.73 | 3.99 | 10 |
| 16.B | 7.2 | 5.15 | 5. | 10 |
| 17.B | 7.1 | 5.14 | 5.14 | 9 |
| 11.B | 6.17 | 3.09 | 3. | 10 |
| 3.A | 6.04 | 2.09 | 3.28 | 9 |
| 1.B | 5.97 | 2.09 | 2.72 | 8 |
| 18.B | 5.7 | 3.27 | 3.43 | 10 |
| 10.B | 4.79 | 1.66 | 2.11 | 9 |
| 4.A | 3.99 | 1.92 | 1.95 | 8 |
| 6.B | 3.84 | 1.44 | 1.72 | 5 |
| 2.A | 3.76 | 1.86 | 2.45 | 8 |
| 12.A | 3.49 | 1.64 | 1. | 7 |
| 14.A | 3.1 | 2.78 | 2.78 | 9 |
| 9.B | 2.99 | 0.92 | 0.92 | 4 |
| 13.A | 2.4 | 0.86 | 0.86 | 7 |
| 7.B | 2.4 | 1.17 | 1.17 | 4 |
| 8.B | 2.37 | 2.05 | 2. | 7 |
| 5.A | 2 | 1.37 | 1.37 | 6 |
| StDev.A | 1.3145 | | 0.842 | |
| StDev.B | 2.0324 | | 1.457 | |
| StDev.AB | 1.9235 | | 1.2913 | |

Table 4: Comparing Linguistic with Cosine Evaluation Measures

sine ones. In Figure 2. we map weighted values by computing the ratio of each sentence level parameters dividing it by the number of masked words to produce normalized versions of the evaluation. In this case we grouped together the A sentences and then the B ones to show their different behaviour.

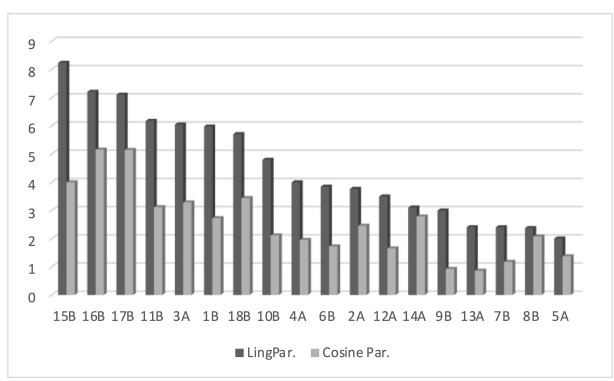

Figure 1: Evaluation by Three Parameters

The predictability score combines the linguistically weighted output of the masked task which is based on embeddings' cosine measure evalua-

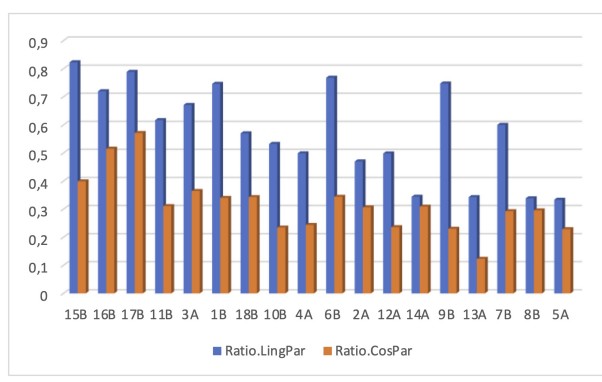

Figure 2: Weighted Evaluation by Two Parameters

tion, and the frequency ranking of each word as reported in the vocabulary list. If we divide up the ratio of the evaluation score by the ratio of the frequency score we obtain the following predictability score: Poetry = 0.561, and News – 4.68. In sum, even though the poetry domain has a smaller number of sentences and almost half the number of words than the newswire domain, the three parameters we evaluated show the correctness of our hypotheses: poetry is by far less predictable. In the poetry domain the two main parameters – word frequency and word context - conspire to reduce the predictability score. The context in poetry domain is characterized by metaphorical usage of word combination accompanied by constituent displacement and discontinuity contributing surprisal effects, but dramatically reducing the ability of BERT to find useful embeddings. These two aspects are further constrained by the low frequency of some words, justifying the low value of the overall predictability parameter. The opposite applies to the news domain: word linear combinations remain fairly literal in their semantic usage thus favouring the possibility for BERT to find good embeddings even when words have low frequency values. As a general remark, then, function words have a much higher cosine score than content words – with the exception of "senatore"/senator and "vita"/life, which being a fairly established collocation or multiword again confirms the relevance of the context, which in the case of function words is the most important parameter to consider.

## 5 Conclusion

In this paper we have proposed a word predictability parameter based on linguistically motivated information that we have tested in a highly constrained context determined by the combination of three fundamental factors for a sentence meaning understanding perspective on the prediction task represented by BERT masked task: use of infrequent words - as measured against the ItWac frequency list - and their phrase level combination – word poetic usage for metaphors w.r.t possible semantic association -, and their larger sentential context in uncommon syntactic structures – noncanonical structures. In order to be able to evaluate the different impact of the three adversarial factors on masked word prediction, we have included in the dataset a higher number of sentences from newswire domain showing the same structural syntactic properties but lacking both the usage of very infrequent words – with a few exceptions - and their uncommon combination to produce metaphors. Word predictability has then been measured by BERT raw word embeddings and their cosine measure, by masking one content word at a time - and a few function words. Each content word has then been searched in the frequency list made available by the ItWac frequencyt list. The results have clearly shown the ability of newswire sentences to receive an overall higher word predictability score thanks to the smaller effect of adversarial factors we investigated. The answer to the question: is frequency or context the determining factor for Transformer Language Models to predict the masked word, is both are. The news domain has less infrequent words and less uncommon noncanonical structures than the poetry domain, which is what explains the remarkable difference in final results.

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

## A  Appendices - English Version of the Canonical and Non-canonical text

1.B Today I thank for the courtesy on several occasions demonstrated to me and my colleagues. 2.A She alone maybe the cold dreamer would educate to the tender prodigy. 3.A I think of a green garden where with you resume can conversing the soul maiden. 4.A If spring my generous heart choked of deaf spasms. 5.A Neither the oblivious enchantment of the hour the iron-like beat grants. 6.B Becomes thus sharper the contradiction. 7.B Good instead overall the rest. 8.B An important decision Ghitti reserved after the holidays. 9.B The important thing is now to open it more. 10.B His information would also give to the guidelines of laique democracy greater boosts. 11.B In this book Maria Teresa, they explain at Mondadori's, will give examples of charities concrete. 12.A Said that they have his heart from inside the chest removed. 13.A The reluctant opinions and not ready and in the midst of executing works hampered. 14.A An echo of mature anguish revverdived to touch signs to the flesh dark of joy. 15.B The government, therefore, though giving up the absolute majority, has wanted, as already in IMI, focusing on a gradual privatization. 16.B At a conference in the Viminale the minister, when he is questioned on the senator to life, at first does not understand the name. 17.B First intervention to do, he said these days, is to implement the reform. 18.B I conceive the private as a work method, as work contracts, as a way to manage in short.

1.Bc Today I thank you for the courtesy demonstrated to me and my colleagues on several occasions. 2.Ac Maybe the cold dreamer educated her alone to the tender prodigy. 3.Ac I think of a green garden where the soul maid can resume conversing with you. 4.Ac Spring if you choked my generous heart of deaf spasms. 5.Ac Neither the iron-like beat of the hour grants the oblivious enchantment. 6.Bc The contradiction becomes thus sharper. 7.Bc Instead, overall the rest is good. 8.Bc Ghitti reserved an important decision after the holidays. 9.Bc Now it's important to open it more. 10.Bc His information would also give greater boosts to the guidelines of laique democracy. 11.Bc In this book Maria Teresa will give concrete examples of charities, they explain at Mondadori's. 12.Ac They said they took off his heart from the chest. 13.Ac The reluctant opinions and not ready works hampered in the middle of executing. 14.Ac An echo of mature anguish revverdressed to touch signs of joy obscure to the flesh. 15.Bc So the government wanted to focus on a gradual privatization while giving up the absolute majority as already in IMI. 16.Bc At a conference in the Viminale, when he is questioned on the senator to life at first the minister does not understand the name. 17.Bc To implement the reform is first intervention to do, he said these days. 18.Bc I conceive the private as a work method, such as work contracts, as a way to manage in short.

## B  Supplemental Material

*Sentence 1.B - Oggi ringrazio della cortesia in più occasioni dimostrata a me e ai miei colleghi.  1.Bc Oggi ringrazio della cortesia dimostrata a me e ai miei colleghi in più occasioni.* The sentence belongs to the newswire domain: it is computed best in the canonical form, with 5 words over 8 while the non-canonical version has only 3 words predicted correctly – only "più/more", "occasioni/chances" and "miei/my". Cosine values are not particularly high except for "miei/my" the possessive which being in its attributive position has a favourable predictive condition. "Oggi" is wrongly predicted as being a separator with very high value, " ‹s› 0.99998". It can be noted that "ringrazio" is partially predicted by "Grazie" in first position but very low value 0.14397. Now the canonical version: Ringrazio (0.0238), più (0.287), occasioni (0.545), dimostrata (0.165), miei (0.882). Interesting to note that the three words predicted in both structural versions have the same cosine values. When we add the remaining 7 sentences, another word is predicted, colleghi (0.076). No connection with frequency values of the missing words: they are all positioned in the high part of the frequency list – excluding "più" and "miei" which are grammatical words and are positioned close to the top. **Frequency List: 5212094-più; 195503-miei; 149546-Oggi; 94921-colleghi; 54876-occasioni; 34756-ringrazio; 15340-dimostrata; 12876-cortesia**

*Sentence 2.A - Lei sola forse il freddo sognatore educherebbe al tenero prodigio.  2.Ac Forse il freddo sognatore educherebbe lei sola al tenero prodigio.* The second sentence belongs to the poetry domain. The original non-canonical version has no candidate found in the first 5 positions. This may be due to presence of a rather infrequent word like "educherebbe/would+educate" as main verb which only appears listed low only in the Upper List. On the contrary, the canonical form has three words predicted: first "Forse/Maybe ", second word "lei/She", and third word "solo"/alone but with wrong masculine morphology. However, these words are correctly predicted with low cosine values - Forse (0.149), lei (0.0355) solo (0.0145). No version provides useful approximations of the meaning of the missing words even though "freddo/cold" is included in the high portion of the 50000 vocabulary. As to the remaining words, they are still included in the Vocabulary but in the lower portion. It is important to note that the lack of prediction can only be motivated just because by combining not so frequent words in unusual combination has produced metaphors like "cold dreamer", "tender prodigy", in association with a verb like "educate". **Frequency List: 2118720-solo; 321303-lei; 117330-Forse; 51970-freddo; *6771-tenero; *3106-prodigio; *2617-sognatore; **13-educherebbe**

*Sentence 3.A - Penso a un verde giardino ove con te*

*riprendere può a conversare l'anima fanciulla. 3.Ac Penso a un verde giardino ove l'anima fanciulla può riprendere a conversare con te.* The non-canonical version of this sentence has two words correctly predicted, giardino/garden, ove/where and a third word with different morphology, in slot 5, Pensa/Think(3rd+person+singular+present+indicative), rather than Penso(1st+person). In the canonical version we find correctly Penso/think in second slot, and another word is added può/can, the modal auxiliary that is now positioned correctly in front of its main verb "riprendere/restart", which is by itself a very frequent verb. As to cosine values, we have the following low values for the canonical version: Penso (0.085), giardino (0.194), ove (0.146), può (0.0865). The non-canonical version has a lower value for Penso but a higher value for giardino (0.291). In the longer context, the interesting fact is constituted by the substitution of "Pensa" with fino/until in the non-canonical version; while in the canonical version Penso/think is moved to a worse position from second slot to last slot, slot 5 and a lower cosine value (0.06112). As to the non-predicted noun modifier "fanciulla/maid", this is certainly an unusual combination even though the two words are highly frequent. The result of the combination is of course a beautiful metaphor which combines "primavera"/spring with "fanciulla"/maid and the garden. Notice the different position of Penso+1st+pers, with respect to Pensa+3rd+pers which is by far less frequent. Now consider the word conversare/conversing which receives the following list of non-word predicted candidates: erare/?? (0.4455), rare/rare?? (0.16737), lare/?? (0.0549), mare/sea?? (0.0479), scere/?? (0.03124). Apart from RARE and MARE which I don't regard being selected for their current meaning but just for being part of the list of subwords, the remaining segments are all meaningless and bear no semantically useful relation with the masked word CONVERSARE. **Frequency List: 1639275-può; 142019-ove; 117242-anima; 102337-verde; 39684-Penso; 32891-riprendere; *9198-Pensa; *8327-fanciulla; *2272-conversare**

*Sentence 4.A - Se primavera il mio cuor generoso soffocasti di spasimi sordi. 4.Ac Primavera, se soffocasti il mio cuor generoso di spasimi sordi.* In this sentence only the phrase "mio cuor"/my heart is predicted in both structural versions. mio (0.291), cuor (0.394). The word "Primavera", which is the first word in the canonical version, has no close prediction: as happens in all sentences, the prediction is totally missed whenever a content word appears in first position. In the non-canonical version, the word comes second, after the conjunction "Se"/If, which predicts the appearance of an auxiliary BE/HAVE in their correct morphological word form – fossi/were, avessi/had in both cases with first person morphology, but also fosse/were, and the last two: con/with and solo/alone. The version with the addition of the 7 sentences has the worsening effect of introducing a subword in place of con/with, MMAI which I assume derives from the wrongly split SEMMAI/if+ever. The word has been wrongly split because the segment SE is wrongly – at least in the word SEMMAI - regarded as a legitimate segment due to its very high frequency. Again the problem seems the unusual combination of the remaining words which are fairly common, apart from soffocasti/choked which is not included in the frequent nor in the Rare wordform list; and spasmi/spasms which is only included in the Upper List. In other words, it's their metaphorical import that prevents the correct prediction. However, it is the position that produces the worst results: the adjective "sordi/deaf" in predicative position is predicted as a punctuation mark in both structural versions. **Frequency List: 762026-Se; 670348-mio; 237398-cuore; 45829-primavera; *9294-generoso; *7333-Primavera; *6503-sordi; **1062-spasmi**

*Sentence 5.A - Né l'oblioso incanto dell'ora il ferreo battito concede. 5.Ac Né il ferreo battito dell'ora concede l'oblioso incanto.* This sentence is the worst case of the poetry domain lot: it has no word predicted neither in the non-canonical nor in the canonical version. This may be due to the presence of a very infrequent word "obliosi/oblivious". However, we notice the presence of an unusual combination of the attributive metaphoric use of "ferreo/iron-like", a rather unusual word. But of course, it is just the combination of words used to build a powerful metaphor that prevents predictions to take place. It is worthwhile noting that "incanto"/enchantment is substituted by ten candidates semantically loosely related to the domains evoked by the masked word: temporal dimension (rhythm, stepping, passing, proceeding, beat), and a condition of the contemplating mind (silence, rest, meaning, thought, sound). Also another important remark regards the inability to predict the ambiguous word "ora"/hour, homograph with "ora"/now, thus clearly showing that context is the determining factor. **Frequency List: 767444-ora; 23438-Né; 15801-concede; 13656-incanto; *5312-battito; **922-ferreo; **14-oblioso**

*Sentence 6.B - Diventa così più acuta la contraddizione. 6.Bc La contraddizione diventa così più acuta.* This sentence has different predicted words in the two structural representations, Diventa/Becomes is present in both. Then "così/so" and "più/more" are predicted in the canonical sentence - diventa (0.215), così (0.0439), più (0.559); while in the non-canonical structure only acuta/sharp is predicted, acuta (0.0441), and the cosine value for "Diventa" is lower being in sentence first position. The canonical form has predicted the discourse marker "così/so" positioned in sentence center: not so in the non-canonical structure where we can again assume that it is the position right after the verb at the beginning of the sentence that does not allow the prediction, notwithstanding its high frequency. Now consider the high frequency of "contraddizione" which is not predicted presumably because of its position at the end of the sentence: the first candidate is the subword "mente" with cosine value (0.16536), followed by sensibilità/sensibility, coscienza/conscience, gioia/joy. **Frequency List: 5212094-più; 1244269-così; 23718-contraddizione; 10807-acuta; *4904-Diventa**

*Sentence 7.B - Buono invece in complesso il resto. 7.Bc Invece in complesso il resto è buono.* No word was predicted in either versions. In order to transform the original non-canonical version in the corresponding canonical one we added the copula "è" that is missing in the original sentence. This is predicted in the canonical version but since it has been added we do not count it for the actual predictive task. All the words are very frequent. As will be clarified further on, whenever the first word of the sentence coincides with a discourse marker or a conjunction the prediction is very close if not equal. This is the case for the canonical form of the sentence starting with "Invece"/Rather, which has the five following best predictions: "Ma"/But, "E"/And, "Però"/However, "Più"/More, "Ed"/And, all belonging to the same grammatical category and in two cases, also to the same semantic type ("Ma", "Però"). Considering the status of the adjective "Buono"/Good which comes in first position in the non-canonical structure and in second position in the canonical one, one can clearly realize the importance of the respective position and the context on the ability of BERT to predict. In the first case, the word coming first position has no left context and there is no similarity, not even at a grammatical level: only conjunctions and verbs are predicted. On the contrary, in the canonical form, "buono" appears as predicate in a copulative structure and the predictions are very close: diverso/different, risolto/resolved, compiuto/achieved, secondario/secondary, positivo/positive. **Frequency List: 736434-invece; 213244-resto; 138658-complesso; *5885-**

**Buono**

*Sentence 8.B - Una decisione importante Ghitti l'ha riservata a dopo le feste. 8.Bc Ghitti ha riservato una decisione importante a dopo le feste.* Only one word is predicted in both versions but it is not the same word. The canonical version predicts "importante/important", (0,0605), the non-canonical version predicts "dopo/after", (0.0152). As can be noticed, the cosine values are very low and again the frequency of occurrence of the words contained in the sentence is fairly high - excluding the proper name "Ghitti" which does not exist in the overall frequency list. The unexpected fact is constituted by the inability to predict the auxiliary "ha"/has in the non-canonical structure – as opposed to what happens in the canonical one -, and the association in fourth slot of a non-word like "vamteen", presumably a subword of some kind. The only explanation could be the presence of a past participle with feminine+singular ending which is only allowed by presence of the resumptive clitic "la" needed to construct the Clitic Left Dislocation of the object NP "Una decisione importante". As said above, the canonical version predicts the presence of the auxiliary HAVE in the correct form and also in two additional morphologically possible forms: "aveva"/had+3rd+pers and "avrebbe"/would+have+3rd+pers; final word predicted in the other auxiliary legal form "è"/is. **Frequency List: 1232332-dopo; 391362-importante; 191762-decisione; 40045-riservata; 30290-feste; Ghitti \*\*\*‹ukn›**

*Sentence 9.B - L'importante ora è aprirlo di più. 9.Bc Ora è importante aprirlo di più.* This sentence is perhaps too short and only function words are captured by BERT embeddings: ora/now (0.3825) più/more (0.0911). The ambiguous word "ora"/now is better predicted in the non-canonical structure - in first position - for the availability of right context - the canonical version predicts "Ora" in fourth position (0.0844). Again this is not relatable to a frequency problem but just structural problems, with the exception perhaps of the final word "aprirlo" which is only present in the very-low frequency list. In fact, in the canonical version, "aprirlo"/open+it is substituted by cliticized verbs - though semantically unrelated, however, showing that the morphology has been captured correctly. As to "importante"/important, it does not appear in the first five candidates, but it is predicted in sixth position (0.04902). **Frequency List: 767444-ora; \*\*1448-aprirlo**

*Sentence 10.B - Le sue informazioni darebbero anche agli orientamenti di democrazia laica maggiori spinte. 10.Bc Le sue informazioni darebbero maggiori spinte anche agli orientamenti di democrazia laica.* This sentence has the same predicted word "maggiori/major" in both structural representations. As before, the words are all very frequent with the exception of "darebbero/+would+give, which is below the threshold and is only part of the "very+low" List. Now consider the word spinte/boosts: predicted masked words are as follows: certezze/certainties (0.0852), garanzie/guarantees (0.0824), informazioni/information (0.04183), taria/tary (0.04003), opportunità/opportunities (0.0383). The fourth slot contains a subword, in fact a non-word, which is assigned a score higher than the one assigned to "opportunities". The question is that the masked word is not frequent enough to be able to collect the co-occurrences required. As a result, even very low scored embeddings are considered. The non-word gets a slightly better score when the text is considered as a whole with the last 7 sentences added, up to (0.06002), but remains always in fourth position. **Frequency List: 4855763-anche; 502931-informazioni; 509780-sue; 157682-maggiori; 130941-democrazia; 24988-orientamenti; 11657-laica; \*9396-spinte; \*1385-darebbero**

*Sentence 11.B - In questo libro Maria Teresa, spiegano alla Mondadori, darà esempi di carità concreti. 11.Bc In questo libro Maria Teresa darà esempi di carità concreti, spiegano alla Mondadori.* In this sentence there is a striking difference in prediction between the two structures. The non-canonical version has only two words predicted, "libro/book" and "esempi/examples", libro (0.0242), esempi (0.653). On the contrary, in the canonical version BERT manages to predict four words, "questo/this", "Maria/Mary", "Teresa/Therese", "esempi/examples", questo (0.767), Maria (0.283), Teresa (0.141), esempi (0.734). Strangely enough, the word "libro" does not figure in the first five candidates. Useless to say, the remaining words are all very frequent. The third run with a longer text including the following 7 sentences gives interesting results: "Teresa" now becomes first candidate substituting the previously chosen first candidate "ci"/us. The word "esempi"/examples, predicted as first candidate, in the text is followed by "carità"/charity which is not predicted in both version: in its place, the first candidate is again "esempi", thus certifying that predictions are made one word at a time disregarding the textual context. Now consider the adjective "concreti" which has been dislocated and is disjoined from its head, "esempi". The list of five candidates for the canonical version is the following: "cristiana+fem+sing"/Christian (0.1919), '.' (0.0909), ',' (0.0387), "civile+sing"/civil (0.0383), "esemplare+sing"/exemplar (0.0222). None of the candidates is plural in number as it should be, if the morphology of Italian has to be respected. On the contrary, the first candidate agrees both in number and gender with the preceding word "carità+fem+sing"/charity, which is not to be considered the correct nominal head. The non-canonical version has one punctuation mark less and an additional adjective "pastorale+sing"/pastoral. **Frequency List: 2980292-questo; 293071-libro; 53531-esempi; 32773-carità; 28289-concreti; 24999-darà; 15537-spiegano; 21854-Mondadori**

*Sentence 12.A - Disse che gli hanno il cor di mezzo il petto tolto. 12.Ac Disse che gli hanno tolto il cuore di mezzo il petto.* This sentence from the poetry subset has only one word in common "cor/heart" and an additional word predicted in the canonical structure, "tolto/taken+off". The cosine values are all very low, cor-cuore (0.1019), for the non-canonical, and cor-cuore (0.0756), tolto (0.156) in the other structure. Interesting enough, when using the configuration with the whole text, also "mezzo/means" is predicted in second slot. **Frequency List: 337473-mezzo; 237398-cuore; 22078-petto; 18406-tolto; \*6176-Disse**

*Sentence 13.A - I ritrosi pareri e le non pronte e in mezzo a l'eseguire opere impedite. 13.Ac I ritrosi pareri e le opere non pronte e impedite in mezzo a l'eseguire.* No prediction found by BERT in the two structural representations - with the exception of "mezzo"/means which however is only appearing in 8th position and not considered in this evlauation. However it is important to note that the previous seven predicted words are in fact only subwords, mostly meaningless, and some having a corresponding identical wordform with a totally different meaning. Here they are: "dotti"/learned+mas+plur, "dotte"/learned+fem+plur, "tente"/meaningless, "sistenti"/meaningless, "sistenza"/meaningless,"difficoltà"/difficulty, "fami"/meaningless. As to their frequency, words are mostly frequent but there are two missing words in the overall frequency lists: "ritrosi/reluctant" and "impedite/hampered". These two words may have been supplemented as subwords but with no useful context for the current analysis. The five candidates appearing are as follows: for "ritrosi" we have - suoi/his+hers, non/not, buoni/good+masc+plur, mal/bad(truncated), loro/their+them+they; and for "im-

pedite" - '.', buone/good+fem+plur, inutili/useless+plur, nuove/new+fem+plur, pubbliche/public+fem+plur. In all of these cases, even if the correct word has not been predicted, the morphology has been matched correctly. **Frequency List: 337473-mezzo; 274709-opere; 43860-pareri; 43387-eseguire; 12619-pronte; \*\*\*ritrosi; \*\*\*impedite**

*Sentence 14.A - Un'eco di mature angosce rinverdiva a toccar segni alla carne oscuri di gioia. 14.Ac Un'eco di mature angosce rinverdiva a toccar segni di gioia oscuri alla carne.* This is another sentence from poetry domain very hard to tackle and to understand. Both the canonical and the non-canonical analyses have just one word found, "eco/echo" (0.0984). Of course the main verb "rinverdiva" is not amongst the frequent words in the list: in fact, it is missing. The remaining words are frequent but they are organized in a peculiar structural configuration with the declared aim to produce metaphors. No changes or improvements when the sentence is analysed with the canonical version of the text. As we did for example 11, we now consider the discontinuous adjective "oscuri+masc+plur"/obscure and the morphology of the five candidates predicted. In the non-canonical version we have: "pieni+mas+plur"/full (0.5461), "piena+fem+sing"/full (0.0486), "e"/and, ',', "pieno+mas+sing"/full (0.0216). Now the canonical version: "fino"/until (0.1139), "intorno"/around (0.1139), "dentro"/inside (0.1001), "sino"/until (0.0476), "vicino"/close (0.0437). As can be noticed, all of the predicted words for the non-canonical structure are function words and none – with the possible exclusion of the ambiguou "vicino+mas+sing" - is an adjective. The reason for this lack of grammatical match may be due to the presence of the articulated preposition "alle"/to the+fem+plur in the canonical version. In the non-canonical version the word "oscuri" was followed by a preposition "di" which is the most frequent wordform with 65 million occurrences. **Frequency List: 6161794-alla; 79244-carne; 64131-gioia; 53363-segni; 21367-toccare; 18431-eco; \*7569-oscuri; \*3490-mature; \*3561-angosce; \*\*\*rinverdiva**

*Sentence 15.B - Il governo, quindi, pur rinunciando alla maggioranza assoluta, ha voluto, come già nell'IMI, puntare a una privatizzazione graduale. 15.Bc Quindi, il governo ha voluto puntare a una privatizzazione graduale pur rinunciando alla maggioranza assoluta come già nell'IMI.* This long sentence belongs to the domain of the news and even in its non-canonical structure, it is more linear and thus more predictable. There are seven words predicted (over ten we masked) in the two versions: governo/government (0.304), maggioranza/majority (0.0377), assoluta/absolute (0.349), ha/has (0.977), voluto/wanted (0.491), puntare/aim (0.0385). The proper name IMI is in the very low list. Strangely enough the function word come/like (0.1925/0.9186) is predicted as first candidate in its non-canonical position, as second position ,but with a much lower cosine measure in canonical position. **Frequency List: 423495-governo; 224791-maggioranza; 126240-voluto; 78651-assoluta; 22290-puntare, 19594-privatizzazione; 18634-graduale; \*5417-rinunciando; \*\*1611-IMI**

*Sentence 16.B - In una conferenza al Viminale il ministro, quando viene interrogato sul senatore a vita, sulle prime non capisce il nome. 16.Bc In una conferenza al Viminale, quando viene interrogato sul senatore a vita sulle prime il ministro non capisce il nome.* There are four words predicted in this long sentence, again in the domain of the news, in the canonical and the non-canonical structures. They are: ministro/minister (0.497), viene (0.795), senatore/senator (0.808), vita/life (0.996). Again, most words are very frequent. An apparent difficulty is constituted by presence of a multiword: "sulle prime/at first" which may be hard to

distinguish and differentiate on the basis of the context. In fact, in both structures, "prime" is substituted by riforme/reforms, banche/banks, dimissioni/resignation , pensioni/pensions, cose/things. **Frequency List: 1099669-vita; 817242-viene: 417438-nome; 228050-ministro; 154970-prime; 104588-senatore: 85517-conferenza; 40680-capisce; \*5529-interrogato; \*5348-Viminale**

*Sentence 17.B - Primo intervento da fare, ha detto in questi giorni, è di attuare la riforma. 17.Bc Primo intervento da fare è di attuare la riforma, ha detto in questi giorni.* This is another fairly simple sentence which has the major number of predicted words in the whole set in relation to the total number in the sentence. There are six words predicted both in the canonical and the non-canonical version: "fare/do" (0.818), "ha/has" (0.283), questi/these (0.961), giorni/days (0.83), riforma/reform (0.194). The only difference being the slot assigned to riforma/reform, which has first slot in the canonical version and second slot in the non-canonical one, preceded by Costituzione/Constitution. Useless to say, the missing words are all very frequent. **Frequency List: 1164126-fare; 782741-giorni; 544195-detto; 354085-intervento; 215894-riforma; 51840-Primo; 35715-attuare**

*Sentence 18.B - Io il privato lo concepisco come un metodo di lavoro, come contratti di lavoro, come modo di gestire insomma. 18.Bc Io concepisco il privato come un metodo di lavoro, come contratti di lavoro, come modo di gestire insomma.* In this final sentence again belonging to the newswire domain, there are four words predicted: metodo/method (0.0618), lavoro/work (0.214), lavoro/work (0.214), modo/way (0.794). Again very frequent missing words, apart from "concepisco/surmise" which is the only word present in the Rare-Words list. When analyzed with the canonical version of the text, the word lavoro/work moves from third to first slot, with a slightly improved cosine score. **Frequency List: 1582948-lavoro; 1111342-modo; 332176-Io; 145442-contratti; 117536-privato; 117677-metodo; 84689-insomma; 70161-gestire; \*\*\*1-concepisco.**