# OpenReview forum: "Word Predictability is Based on Context – and/or Frequency"
_aclweb.org/ACL/2022/Workshop/CMCL — Submitted to CMCL 2022_

### Official Review · Reviewer_P5MN · 2022-03-23
**Interesting work but lacking in many aspects**

**Rating:** 3
**Confidence:** 5

**Review:**

The current work investigates word prediction in Italian sentences. A BERT model is used to predict a masked word in sentences with differing complexity. The complexity of the sentences is operationalised via the genre of the text (new vs poetry); in particular via frequency of the word as well as word order/semantics.

While this is an interesting and pertinent investigation, I am afraid the way the paper is written makes it extremely difficult to understand the experimental setup, the key hypothesis and the results. For example, at no point in the paper do the author clearly state how the masked word prediction task is operationalized. The authors state "To this aim we ran BERT by masking each content word and some function word, ... " -- what does masking some function word mean? Relatedly, it is unclear if while making a prediction, the model has access to the right context of the masked word. This is important because the model is being compared to how humans predict a word. Human word prediction is incremental in nature where only the left context is used to make upcoming prediction.

In addition, the work is not contextualised in the larger psycholinguistic prediction literature; the current citations in section 2 is cursory at best.

My suggestion would be that the authors should completely rewrite the paper by making the research questions, the experimental method and the results more accessible.

Finally, the submission to the CMCL workshop has to be anonymous and self citations have to be avoided -- the authors write "We already discussed elsewhere (Delmonte, 2021) that languages like Italian, which have a rich morphology, ..."; such a statement is a clear violation of the guidelines.

---

### Official Review · Reviewer_DaR1 · 2022-03-23
**Linguistically relevant but computationally vague work**

**Rating:** 3
**Confidence:** 5

**Review:**

This paper explores the ability of Transformer Models to predict a word in Italian sentences. The authors investigate how predictions vary in canonical and non-canonical order of the same sentences from two different domains (poetry and newspapers).

The investigation is relevant, and the linguistic motivations are well defined. However, there are some substantial limits.
On a theoretical side, the authors claim their work is part of the line of research where human word predictivity is compared and tested by the performance of DNNs in next-word-prediction tasks. If this is was the aim, authors should test their results with behavioral data. In my opinion, this is more of a study on the linguistic abilities of Transformer Models.

Moreover, the dataset selected is too small to make strong claims about the results. While it is true that this allows a detailed error analysis, the setup is questionable; among all, sentences are not balanced between the two domains.

The critical aspect, however, is the computational implementation. The description of the experimental setup is too vague.
On the one side, some decisions are not motivated (why do they choose the output of the first or projection layer and not other layers?), and the resources used are not clearly stated. For instance, the authors state they use BERT, but then they cite UmBERTo, which actually inherits from RoBERTa the base model architecture. On the other side, the way the evaluation is described is so obscure that it is hard to understand what was concretely done, yet making it hard to reproduce. For instance, it is unclear what they mean by 'We evaluate word co-occurrence frequencies by means of embeddings as the cosine value made available by BERT': the cosine is used as a measure to compute the similarity between two word-embeddings, but in the paper, I cannot find which are the two embeddings directly compared. The prediction technique should be given more details, specifically how the authors deal with cases when a word is split into subtokens by the tokenizer.

I recommend that the authors rewrite the paper entirely by reformulating the research questions and detailing the experimental method precisely.

Last but not least, the paper is not anonymized. Indeed, in two cases, the authors explicitly refer to their previous works (line 126: 'We already discussed 126 elsewhere (Delmonte, 2021)..',  and footnote 5 '5We comment and analyze in depth all sentences in a paper..'), which is against the double-blind reviewing policy.

---

### Official Review · Reviewer_YeEh · 2022-03-24
**Not relevant for the CMCL venue**

**Rating:** 2
**Confidence:** 4

**Review:**

The paper compares BERT predictions for texts written in two different registers, in Italian: newspapers and poetry.

I do not see a fit for this paper in this venue, as there is no cognitive modelling involved, just a test of BERT's predictions for different registers. There is a section reviewing a small amount of computational psycholinguistic work in sentence processing, but the current study differs from the reviewed ones in that it does not compare against human behavior. Hence conclusions cannot be related to cognitive processing in any meaningful way.

The dataset is extremely small (18 sentences). The authors justify this choice based on the fact that they can do more extensive qualitative analysis on a small dataset. However, given this size, it is highly doubtful that the results are robust enough to draw any conclusions.

Other comments:
- acronym DL is not introduced in the abstract, and it seems like a too general way to refer to the models used in the paper, which are all instances of BERT, as far as I could tell.
- explaining what is non-canonicity in the introduction would be helpful
- self-citations should be avoided to preserve anonymity

---

### Decision · Program_Chairs · 2022-03-29

Reject